# A Multicentre Evaluation of Dosiomics Features Reproducibility, Stability and Sensitivity

**DOI:** 10.3390/cancers13153835

**Published:** 2021-07-30

**Authors:** Lorenzo Placidi, Eliana Gioscio, Cristina Garibaldi, Tiziana Rancati, Annarita Fanizzi, Davide Maestri, Raffaella Massafra, Enrico Menghi, Alfredo Mirandola, Giacomo Reggiori, Roberto Sghedoni, Pasquale Tamborra, Stefania Comi, Jacopo Lenkowicz, Luca Boldrini, Michele Avanzo

**Affiliations:** 1Fondazione Policlinico Universitario “A. Gemelli” IRCCS, 00168 Roma, Italy; jacopo.lenkowicz@policlinicogemelli.it (J.L.); luca.boldrini@policlinicogemelli.it (L.B.); 2Prostate Cancer Program, Fondazione IRCCS Istituto Nazionale Tumori, 20133 Milano, Italy; Eliana.Gioscio@istitutotumori.mi.it (E.G.); Tiziana.Rancati@istitutotumori.mi.it (T.R.); 3IEO, Unit of Radiation Research, European Institute of Oncology IRCCS, 20141 Milan, Italy; cristina.garibaldi@ieo.it; 4I.R.C.C.S. Istituto Tumori “Giovanni Paolo II”, 70124 Bari, Italy; a.fanizzi@oncologico.bari.it (A.F.); r.massafra@oncologico.bari.it (R.M.); p.tamborra@oncologico.bari.it (P.T.); 5Fondazione CNAO, Strada Campeggi 53, 27100 Pavia, Italy; Davide.Maestri@Cnao.it (D.M.); Alfredo.Mirandola@Cnao.it (A.M.); 6IRCCS Istituto Romagnolo per lo Studio dei Tumori (IRST) “Dino Amadori”, 47014 Meldola, Italy; enrico.menghi@irst.emr.it; 7Medical Physics Service, Radiation Oncology Department, Humanitas Clinical and Research Hospital, 20089 Rozzano-Milan, Italy; giacomo.reggiori@humanitas.it; 8Azienda USL-IRCCS di Reggio Emilia, 42122 Reggio Emilia, Italy; Roberto.Sghedoni@ausl.re.it; 9IEO, Unit of Medical Physics, European Institute of Oncology IRCCS, 20141 Milan, Italy; stefania.comi@ieo.it; 10Medical Physics Department, Centro di Riferimento Oncologico di Aviano (CRO) IRCCS, 33081 Aviano, Italy; mavanzo@cro.it

**Keywords:** dosiomics, dose distribution texture analysis, multicentric study, reproducibility, stability, sensitivity, radiation dosimetry, radiotherapy

## Abstract

**Simple Summary:**

Dosiomics is born directly as an extension of radiomics: it entails extracting features from the patients’ three-dimensional (3D) radiotherapy dose distribution rather than from conventional medical images to obtain specific spatial and statistical information. Dosiomic studies, in a multicentre setting, require assessing the features’ stability to dose calculation settings and the features’ capability in distinguishing different dose distributions. This study provides the first multicentre evaluation of the dosiomic features in terms of reproducibility, stability and sensitivity across various dose distributions obtained from multiple technologies and techniques and considering different dose calculation algorithms of TPS and two different resolutions of the dose grid. Harmonisation strategies to account for a possible variation in the dose distribution due to these confounding factors should be adopted when investigating a correlation between dosiomic features and clinical outcomes in multicentre studies.

**Abstract:**

Dosiomics is a texture analysis method to produce dose features that encode the spatial 3D distribution of radiotherapy dose. Dosiomic studies, in a multicentre setting, require assessing the features’ stability to dose calculation settings and the features’ capability in distinguishing different dose distributions. Dose distributions were generated by eight Italian centres on a shared image dataset acquired on a dedicated phantom. Treatment planning protocols, in terms of planning target volume coverage and dose–volume constraints to the organs at risk, were shared among the centres to produce comparable dose distributions for measuring reproducibility/stability and sensitivity of dosiomic features. In addition, coefficient of variation (CV) was employed to evaluate the dosiomic features’ variation. We extracted 38,160 features from 30 different dose distributions from six regions of interest, grouped by four features’ families. A selected group of features (CV < 3 for the reproducibility/stability studies, CV > 1 for the sensitivity studies) were identified to support future multicentre studies, assuring both stable features when dose distributions variation is minimal and sensitive features when dose distribution variations need to be clearly identified. Dosiomic is a promising tool that could support multicentre studies, especially for predictive models, and encode the spatial and statistical characteristics of the 3D dose distribution.

## 1. Introduction

In the era of personalised medicine and targeted therapy, one of the most promising methods introduced in clinical practice is radiomics [1]. The key idea behind radiomics is that we can mine images by extracting image descriptors, called radiomic features, which can provide rich information about the tumour or healthy tissue and can be used to build predictive or prognostic models. This method allows quantitative analysis of different

Image modalities and identification of patterns and correlations among voxels that can be of interest for improving diagnosis, prognosis and prediction of treatment outcomes [2,3,4]. Clinical outcomes can be therefore predicted employing radiomics features, potentially changing the treatment paradigm. Nevertheless, several studies highlight the importance of providing robust and unbiased descriptors. Objective quantification of reproducibility, stability and redundancy of features is a prerequisite for radiomics. This kind of process has been performed widely in radiomics [5,6,7,8,9,10,11,12], and it is even more meaningful when performed in a multicentric setting [13,14]. Dosiomics is born directly as an extension of radiomics; it entails extracting features from the patients’ three-dimensional (3D) radiotherapy dose distribution rather than from conventional medical images [15,16] to obtain specific spatial and statistical information. Furthermore, it can parameterise the dose distribution in particular regions of interest (ROIs) by intensity, textural and shape-based features allowing the description of the dose distribution at a high complexity level, distinct from those obtained from dose–volume histograms (DVHs) [17]. Indeed, 3D dose distribution optimisation and evaluation are still mostly based on DVH endpoints, dose distribution visual inspection and DVH-based metrics. Nevertheless, the well-known drawback of DVH is to collapse the 3D dose information in 2D metric, losing the information on its spatial and statistical distribution. The integration of dosiomics with the DVH could constitute an advanced tool to evaluate the radiotherapy plan quality [18] by identifying new dose distribution metrics based on dosiomic features. A second appealing development is introducing dosiomic features into Tumour Control Probability (TCP) and Normal Tissue Complication Probability (NTCP) models, thus overcoming the current limitation of these models [19]. Some authors recently employed dosiomics to improve the prediction of side effects [20,21,22] or local control after radiotherapy [23], including preliminary multicentre experiences [24]. The proposed dosiomic signatures must be highly stable and reproducible and need validation before being used in clinical practice. Developing robust models requires ample training and validation datasets with radiotherapy data from many patients for any specific cancer site. These needs settle dosiomics in the framework of “big data” and push towards multicentre studies. Possible sources of variation for radiomic features include different radiotherapy techniques, treatment planning systems (TPSs), dose calculation algorithms and dose grid resolutions. The variability due to these sources may hide any potential variability associated with the dose–response, making at least some of the dosiomic models unreliable and preventing the generalization of results. In this frame, Placidi et al. evaluated the robustness of dosiomic signatures across grid resolution and algorithm for dose calculation [25] in a monocentric setting. The results of that study highlighted the not negligible variation in dosiomic features, especially for target region and for dosiomic textural features; therefore, dosiomic studies should always provide a reporting of grid resolution and algorithm dose calculation. We here propose to investigate the stability of dosiomic features in a multicentre setting with two main aims: (a) to provide an assessment of the stability of dosiomic features to dose calculation settings and (b) to assess the dosiomic features capability in discriminating dose distributions that are generated with different radiation therapy devices. This study provides the first multicentre evaluation of the dosiomic features in terms of reproducibility, stability and sensitivity across various dose distributions obtained from multiple technologies and techniques and considering different dose calculation algorithms of TPS and two different resolutions of the dose grid.

## 2. Materials and Methods

The evaluation of dosiomic features’ extraction from different dose distributions was performed by several centres, which participate in the Dosiomics Team of the Radiomics Working Group of “Alliance Against Cancer” (Alleanza Contro il Cancro, ACC), a national oncology network founded in 2002 by the Italian Ministry of Health. Specifically, nine centres have contributed to this analysis. Dose distributions were generated by eight out of the nine centres involved in the study on an image dataset acquired on a dedicated phantom (see the specific section below) shared among the centres. Each of these centres could provide more than one dose distribution based on the availability of technologies and delivery techniques.

### 2.1. Phantom

A computed tomography (CT) scan of a cylindrical heterogeneous phantom was acquired for treatment planning and dose calculation. In particular, the ArcCheck [26] PMMA insert (ArcCHECK MR Sun Nuclear Melbourne Florida, US) was modified by substituting 4 PMMA rectangular sub-inserts with the following equivalent densities: lung, bone, muscle and adipose. The planning CT (GE. Optima CT580 W HiSpeed DX/I Spiral) had a slice thickness of 1.25 mm, 140 kV, pixel size of 1.269 mm^2^, as shown in Figure 1.

On the acquired planning CT, the regions of interest (ROIs) simulating the tumour and organs at risk (OARs) of a head and neck radiotherapy treatment were contoured. These included a planning target volume (PTV), left parotid, right parotid, spinal canal, planning organ-at-risk volume for the spinal canal (PRV, i.e., isotropic 4 mm expansion of spinal canal) and trachea. Moreover, we added a RING structure defined as an expansion of 3 cm from the PTV and cropped of 0.0 cm from the PTV edge to ensure a high dose gradient outside the PTV. The planning CT and its ROIs were exported in DICOM format and shared among the centres. In terms of PTV coverage and dose–volume constraints to the OARs, two different planning protocols, including minimum and maximum dose to the PTV and dose–volume constraints to the OARs, were followed by participants to produce comparable dose distributions, which were used to evaluate dosiomic features in terms of reproducibility, stability and sensitivity. Dose distribution computation was performed from eight different centres, named A, B, C, D, E, F, G, H.

### 2.2. Plan and Dose Prescription: Same Techniques, Technology and TPS

To evaluate the reproducibility and stability of dosiomic features, we planned a series of Intensity Modulated Radiation Therapy (IMRT) treatments with dose distributions as equivalent as possible, employing the same delivery technique, a unique dose distribution optimisation protocol and identical or similar LINACs. With reproducibility, we mean that a result obtained by an experiment should be achieved again with a high degree of agreement when the study is replicated with the same methodology by different researchers. A stable measure, on the other hand, is one in which the sources of variation are consistent over different inputs and conditions; here it is TPS and Technologies. This means that the process does not exhibit unpredictable variation for this purpose. We chose almost similar LINACs and the same photon energy, gantry angles, TPS and planning objectives. We considered only IMRT 6MV FF Varian machines (Trilogy, TrueBeam, TrueBeam Edge and Clinac) with the Eclipse-Aria TPS for this study phase. The normal tissue objective (NTO) tool was employed with the default setting, and dose grid resolution (optimisation and calculation) was set to 1 mm. Table 1 reports the details of the IMRT protocol.

Eight IMRT dose distributions provided by different centres were included in the “stability” dataset. Table 2 summarises the Varian (Varian Medical Systems) Linacs used, while all the other plan parameters, equal for all the centres, are shown in Table 1.

Figure 2 shows the eight dose distributions included in the studies. A reproducibility study (smaller green rectangle) was conducted on the dose distribution obtained by plans E_1, B_1, D_1 and A_1 (all TrueBeam Linac), while stability study (red rectangle) includes all eight dose distributions listed in Table 2.

### 2.3. Plan and Dose Prescription: Different Techniques, Technologies and TPSs

To evaluate the sensitivity of the dosiomic features extraction to different techniques, technologies and TPSs, each centre planned one or more treatments using a range of different technologies among those available to the centres involved. Sensitivity is defined as the smallest absolute amount of change that can be detected by a measurement. The different delivery techniques, accelerators, TPS and dose calculation algorithms considered in this study are reported in Table 3.

Eleven dose distributions provided by different centres were included in the dataset. Each dose distribution was calculated and optimised with two different dose grid resolutions: 1 mm and 2 mm, always keeping the dose to PTV and the OARs within prescription and constraints. No limitation was imposed in terms of beam setup and geometry. The dose prescription simulated a theoretical head and neck mono-lateral treatment plan with a prescribed dose to the PTV of 66 Gy and dose per fraction of 2.2 Gy. Dose prescription and OARs constraints are summarised in Table 4. Figure 3 shows the resulting eleven dose distributions with 1 mm dose grid resolution.

### 2.4. Extraction of Dosiomic Features

The extraction of dosiomic features was centralised and carried out by a specific routine in the MODDICOM library, a free software package developed in R language optimised for automatic loading of DICOM images and radiomic analysis [28]. A specific routine for dose distribution texture analysis was realised for the purpose of this study, loading and processing the required DICOM dataset (planning CT, RT-Structure and RT-Dose). The features definition, nomenclature and extraction methodology following the one used for radiomic studies based on medical images, as accurately described by Zwanenburg et al. [29]. In dosiomics, the “image” is constituted by voxels with their grey level corresponding to the absolute dose in Gy. The absolute dose levels were binned in 100 discrete levels from zero to max dose before performing feature extraction. A total of 212 dosiomics features defined in the Image Biomarker Standardisation Initiative (IBSI) [29] were extracted from the selected ROIs (PTV, left parotid, right parotid, spinal canal, trachea and RING) belonging to the following families: 17 intensity-based statistics (STAT), 100 features from grey level co-occurrence matrix (GLCM), 63 from grey level run length matrix (GLRLM) and 32 from grey level size zone matrix (GLSZM). Morphological features were not included in this study since not considered relevant for dosiomic analysis.

### 2.5. Data Analysis

The analysis mirrored the two main goals of the study: to assess (a) the stability of dosiomic features to dose calculation settings and (b) the sensitivity to a change in dose distribution, that is, the ability of dosiomic features in distinguishing dose distributions generated with different radiation therapy devices.

As a preliminary test, we evaluated the software reproducibility for the computation of dosiomic features by extracting them in two different centres, both employing the MODDICOM library. We considered the complete set of 212 dosiomic features extracted from the same dose distribution (from centres A) computed with 1 mm and 2 mm calculation grid resolution and from all the contoured ROIs for this check. Differences in values for single features were then analysed. The expected differences are zero since dosiomic features extraction should not depend on the same software employed in different centres. 

We used the coefficient of variation (CV) to evaluate the stability of the dosiomic features to dose calculation settings, i.e., when dosiomic features are derived from equivalent plans (“IMRT-Linac reproducibility”). The CV is a standardised measure of the dispersion of a distribution leading to the degree of intra-features variability. It is defined as the ratio of the standard deviation σ concerning the mean value μ (or to its absolute value |μ|). CV, to respect to standard deviation, is recommended when datasets with different units or widely different means were considered.

For assessing reproducibility, we computed the CV for the dosiomic features extracted by four IMRT dose distributions with the same technique, technology, TPS and Linac version (Varian, TrueBeam, Eclipse, AAA). This analysis investigates reproducibility as the features are extracted after the experiment (here, dose calculation) was replicated with the same methodology (here, same plan, same TPS, same LINAC) by different researchers (here, different centres). All ROIs were considered individually.

In the stability analysis, we still computed CV among the dosiomic features extracted by the entire set of eight IMRT dose distributions derived from the same technique, technology and TPS version. In this case, we considered different Linac Technologies (see Table 4). This analysis investigates stability as features are considered for their possible variation over different inputs and conditions (here, different TPSs and LINACs) to prove that the process does not exhibit unpredictable variation. Awareness and quantification of these variations should always be taken into account to avoid misinterpretation of results from studies, including dosiomic features. 

The sensitivity of dosiomic features to dose distributions generated with different radiation therapy devices, TPSs and algorithms was also evaluated in terms of CV (see Table 5). In this case, CV describes how a dosiomic feature can change due to different dose distributions due to different techniques, technologies, TPSs, dose calculation algorithms, energies and beam quality. 

We adopted a common guideline of thresholding CV value as a strategy to select stable, reproducible and sensitive features. With a view to future multicentric studies concerning tumour control and/or OARs toxicity, it is crucial to select dosiomic features with a small CV, for example, with a CV < 0.3, when dose distributions are expected to be stable and reproducible. Simultaneously, it would also be desirable to identify dosiomic features able to recognise true differences in the dose distributions, so with a large CV, e.g., with a CV > 1, which classifies the sensitivity of the dosiomic features to the dose distribution variation. Since the proposed threshold values are a completely arbitrary choice, the CV > 0.8 threshold was also employed to investigate further and evaluate the variation in the dosiomic feature’s sensitivity on the selected threshold. Dosiomic features that are both stable (CV < 0.3) and sensitive (CV > 1 or CV > 0.8), i.e., that constitute an optimum set for modelling purposes, were described through Venn diagrams. 

## 3. Results

We extracted a total amount of 38,160 dosiomic features from 30 different dose distributions from six ROIs, grouped by four features’ families. In terms of reproducibility of dosiomic features extraction using the same software, two centres extracted 212 dosiomic features for each calculation grid resolution size (1 mm and 2 mm), resulting in a comparison of 424 features between the centres. Single feature value differences, both for 1 mm and 2 mm calculation grid resolutions, were found to be equal to zero for all the considered dosiomic features. This result confirms the reproducibility of the dosiomic features when extracted using the MODDICOM software package (version 0.52).

We evaluated 5088 and 10,176 dosiomic features to assess the reproducibility and stability of the extracted dosiomic features, respectively. Appendix A show the CV values grouped by ROIs and features’ family for reproducibility and stability studies, respectively. 

Concerning the sensitivity study, we extracted 27,984 dosiomic features from the entire set of 11 dose distributions and six ROIs. Appendix A depict the CV values grouped by ROIs and feature’ family, respectively, for the 1 mm and 2 mm dose grid calculation sensitivity studies. 

Results are also summarised in Figure 4 and Figure 5 in terms of box plots for the left parotid and PTV, respectively. The black horizontal line within the box display for each box the median CV value. All the other ROIs (right parotid, spinal canal, trachea and RING) are reported in Appendix A. Additionally, Appendix A lists the mean CV values for all the studies, ROIs and family’s features. 

The Venn diagrams in Figure 6 highlight the features that are both stable and sensitive after the choice of specific CV threshold (CV_TH_) values. For example, results for the set of stable AND sensitive features for the ROIs PTV and left parotid are given in Figure 3 for two different CV_TH_ values for the sensitivity, CV > 1 and CV > 0.8, while the CV threshold for stability is kept to 0.3.

The Appendix A report Venn diagrams highlighting dosiomic features that are both stable and sensitive for the other ROIs considered in this analysis. Table 5 summarises the percentage of features that are both stable and sensitive (CV threshold = 1 and different resolution of the grid for dose calculation, 1 mm vs. 2 mm) across all the ROIs, and the details are shown in Appendix A.

## 4. Discussion

Dosiomic is increasingly used in clinical studies aiming to improve the prediction of clinical outcomes, e.g., locoregional recurrence after IMRT for head and neck cancer [23] or local control after carbon-ion radiotherapy in skull-base chordoma [21]. Dosiomic features were analysed by machine learning for the prediction of acute-phase weight loss in lung cancer patients treated with radiotherapy [30]. Among preliminary multicentre experiences, Adachi et al. [24] aimed at predicting radiation pneumonitis after lung stereotactic body radiation therapy using dosiomics. In both single and, especially, multicentre studies, consistent reporting of dose distribution to provide a robust setting for the study is a key point to ensure stronger validation of the use of dosiomic features in the clinical routine.

The presented study provided the first assessment of the variation in dosiomic features to dose calculation environments in a multicentre setting and the sensitivity of dosiomic features in distinguishing dose distributions generated with different radiation therapy devices.

If considered reproducibility and stability studies, dosiomic features’ families with higher mean CV are always SZM apart from the RING and left parotid ROIs where STAT family shows the higher CV mean value. In terms of capability in describing and evaluating 3D dose distribution to a higher level than DVHs and employing dosiomic features in predictive modelling, CV mean values could not provide any useful information. Nevertheless, this study can describe a peculiar behaviour of the single dosiomic features (listed in Appendix A) and families that could be representative for further studies.

The box plots in Figure 1 and Figure 2 and Appendix A, highlight how dosiomic features depend on dose distribution, ROIs and feature families. As expected, larger CV variations were observed in the sensitivity studies (with a dose calculation grid of 1 mm and 2 mm). It is difficult to generalise these results due to the dose distribution dependency, but, in terms of ROIs, it is visible that ROIs that lay in the gradient region (RING and right parotid) show lower CV values on average (Appendix A). Concerning the dosiomic features’ families, GLCM has almost always the lowest value for all the studies and all the ROIs except for the right parotid in the sensitivity study using a dose calculation grid of 1 mm, PTV and spinal canal in the sensitivity study using a dose calculation grid of 2 mm. Even though the mean value of a single feature is considered, these results highlight how the GLCM features families show the lowest variation in terms of CV. Dosiomic features’ families with higher mean CVs in the sensitivity studies (both with1 dose calculation grid of 1 mm and 2 mm) are RLM and SZM in the 87.5% of the cases for the RING, left parotid, right parotid and PTV ROIs, while STAT for spinal canal and trachea.

Of note, reproducibility and stability of features can also be evaluated in terms of intraclass correlation coefficient (ICC), as is customary in most studies in which radiomic and dosiomic characteristics are evaluated. ICC is defined as the ratio of the subject variance by the sum of the subject variance, the rater variance and the residual, where a lower rater variance implies a reliable scale. ICC expresses how strongly the components in the same group resemble each other [10]. The peculiar analysis presented in this study forces the TPSs/Linacs/RT techniques/RT Technologies to be the “raters” of the dose distributions, while the different ROIs would be the “subjects”. Nevertheless, the ROIs present with very different dose distributions (high doses vs. low doses, almost uniform dose distribution vs. high gradient dose distribution), which means high variation between-subjects (possibly larger than variation among raters) that could lead to biases in ICC calculations. For these reasons, we chose to stick to the coefficient of variation that does not require an evaluation across different “subjects” and only needs evaluation across different “raters”. Nevertheless, an example of ICC evaluation is reported in the Appendix A considering the STAT dosiomic features’ family and the three possible ROIs groups: all ROIs, high dose region ROIs and low dose region ROIs.

The main aim of this study was to provide suggestions on a set of dosiomic features that are at the same time reproducible, stable and sensitive, i.e., robust across variations that are not related to true differences in the dose distribution and able to pick up even subtle true differences in the dose distributions. To achieve this result, a specific guideline of thresholding CV values was defined to filter out reproducible, stable and sensitive features.

The choice of CV thresholds, i.e., CV < 0.3 to define reproducibility and stability and CV > 1, or CV > 0.8, to define sensitivity, is somehow arbitrary. To date, there are no specific and shared reference threshold values; our choice was driven by some statistical considerations. A CV < 0.3 means that the standard deviation of the value distribution for the single feature is less than 30% of the mean value of the same distribution, which means a reasonably low variation across the distribution, with 68% of values in the interval “mean value ±30%”. A CV > 1 (or > 0.8) means that the standard deviation of the value distribution for the single feature is (almost) the same “size” as the mean value, which entails a high possibility that values sampled from such a distribution can be identified as significantly different after a statistical test, which would be desired in outcome modelling. 

This approach allowed identification of dosiomic features that are both stable and sensitive, as depicted in Figure 3 for PTV and left parotid, summarised in Table 1 for all the ROIs and dosiomic features’ families and detailed in Appendix A for all the dosiomic features. As an example, identifying features that overlap between sensitivity study with 1 mm and 2 mm dose calculation grid is potential information that could be useful retrospective multicentre studies where different dose grid resolutions were employed, or for prospective studies to evaluate the possible need of guidelines on the dose calculation grid settings.

We would like to emphasise once more that the CV threshold values selected to filter out and define reproducible, stable and sensitive dosiomic features are not absolute suggested values to take as completely “a priori” reference in future dosiomics studies. Our choice was grounded on some statistical considerations, and results are possibly associated with the peculiar nature of our study, i.e., a phantom study, fixed centralised contouring of ROIs, common dose calculation protocol including fixed-dose prescription, planning objectives and OARs constraints. Other thresholds on CVs could be selected for other studies considering different features distributions, e.g., a possible clear bimodal distribution which derives from “merging” of two separated distributions for patients with/without a selected clinical outcome. 

The employment of dosiomic in clinical practice could represent a powerful tool to handle better the 3D dose spatial and statistical information if compared with conventional tools, such as DVH and DVH metrics. Potentially, the granularity and quantity of the information provided by the dosiomic features, and above all the usability of such information, could better support the clinical decision than standard parameters, such as DVH, DVH metrics and visually assessment of the 3D dose distribution. Obviously, what is still needed is a clinical translation of the meaning of each dosiomic feature both within the use of full 3D dose distribution as a new metric to better assess plan quality during the optimisation phase as well as during the plan evaluation, but also to finally include dosiomics in the predictive models. Dosiomic features could represent additional parameters to be employed in the predictive models: this could lead to identifying some disomic features that, both during the plan optimisation and evaluation, could be considered to prevent or limit acute toxicities, as well as to improve local control.

Additionally, to exploit the full benefits of big data, machine and deep learning, multicentre trials are needed [31]. Multicentre studies (both retrospective and prospective) are strongly based on the quality of the selected parameters. How do we best use the parameters we have been using so far? Are there any other parameters that could support future studies? Dosiomics features could be one of these, being potentially much more sensitive to dose distribution variation. Moreover, in a multicentre trial, a priori selection of the optimal dosiomic features to be employed in the study would lead to more robust and unbiased studies. According to the present analysis, it is essential to highlight how even just the variation in different dosiomic features underlines a possible use of the dosiomics to select dose distribution within multicentric studies to avoid bias during the further clinical outcome correlation analysis.

The first limitation of the present study is related to the pool of the considered radiotherapy techniques and technologies. They are pretty diverse and representative but do not describe all the possible techniques and technologies available in clinical practice. Despite this, we believe that the employed number of radiotherapy techniques and technologies used by the eight centres are enough to support the message that a substantial number of dosiomic features are stable, and at the same time, they can distinguish or recognise dose distributions generated with different radiation therapy devices. 

A second possible limitation is related to the number of considered features. We considered 212 dosiomic features, other dosiomic features of the second-order could indeed have been considered. Nevertheless, the selected features represent a robust dataset, internationally validated in the radiomics setting [29], available to proceed in the clinical implementation of the dosiomics, both in clinical outcome predictive models and in the 3D dose distribution description, optimisation and evaluation processes.

As a further study, dosiomic feature extraction analysis on different software [32] should also be considered to evaluate the possibility of employing different software to extract dosiomic features in multi-institutional studies.

## 5. Conclusions

The present study has assessed the stability of dosiomic features and their capability in distinguishing dose distributions generated with different radiation therapy devices in a multicentre setting. These results suggest that being dosiomic features sensitive to changes in dose calculation parameters, a consistent reporting of the TPS, dose calculation algorithms and pixel spacing used to calculate dose distributions is required. Harmonisation strategies to account for a possible variation in the dose distribution due to these confounding factors should be adopted when investigating a correlation between dosiomic features and clinical outcomes in multicentre studies.

## Figures and Tables

**Figure 1 cancers-13-03835-f001:**
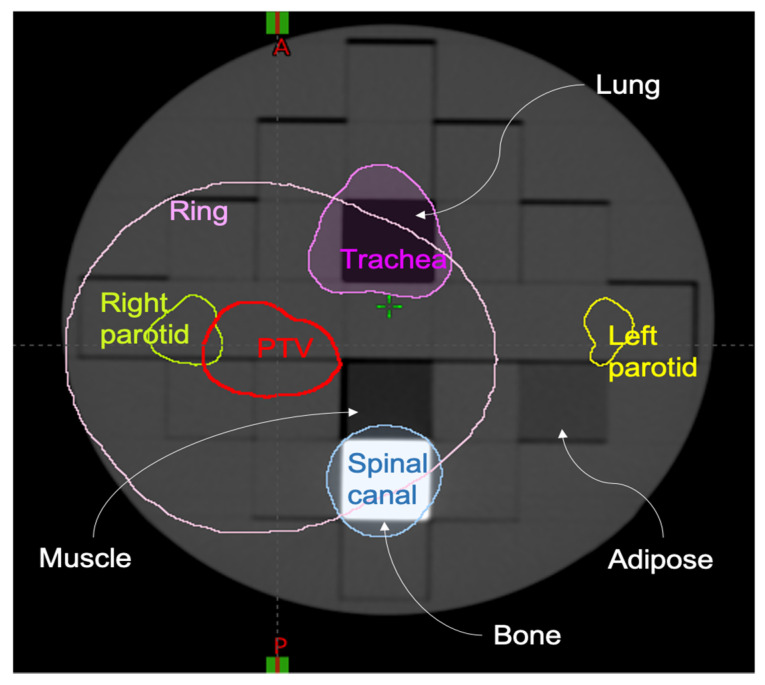
Planning CT of the phantom employed in the study and the countered Regions of Interest (ROIs). Six different ROIs were contoured: planning target volume (PTV), left parotid, right parotid, spinal canal, trachea and RING. Ring structure is the expansion of 3 cm from the PTV and cropped of 0.0 cm from the PTV edge.

**Figure 2 cancers-13-03835-f002:**
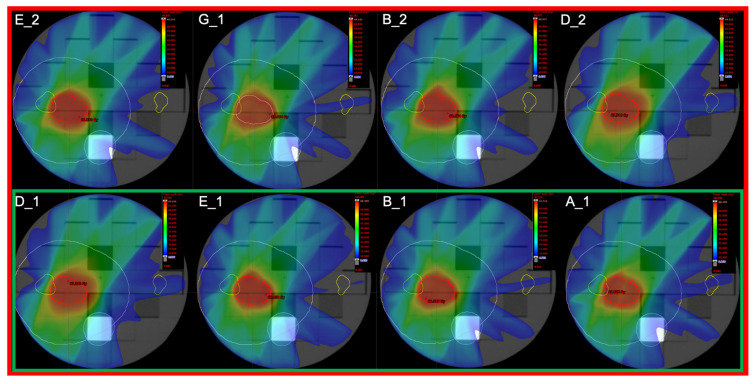
The four IMRT dose distributions included in the reproducibility study (within the green rectangle), and the eight IMRT dose distributions included in the stability study (within the red rectangle).

**Figure 3 cancers-13-03835-f003:**
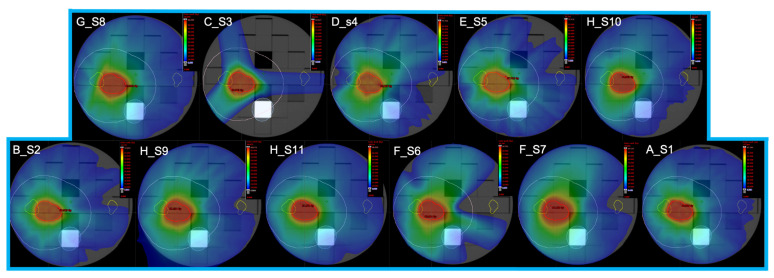
The eleven dose distributions obtained by different techniques, technologies and treatment planning systems, with 1 mm dose grid resolutions.

**Figure 4 cancers-13-03835-f004:**
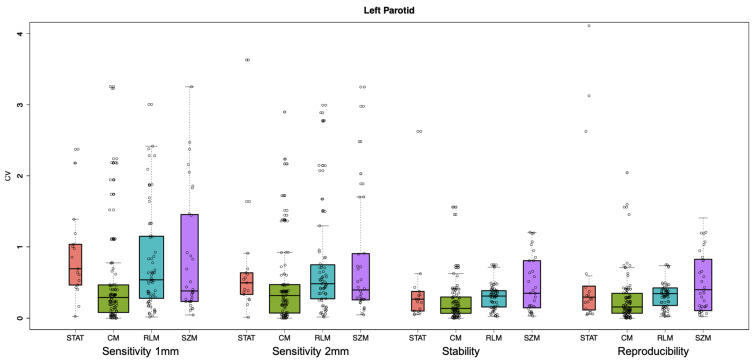
Box plots of the CV values for the sensitivity (1 mm and 2 mm), stability and reproducibility studies, grouped for the four different features’ families (STAT, CM, RLM and GSZ) for the left parotid.

**Figure 5 cancers-13-03835-f005:**
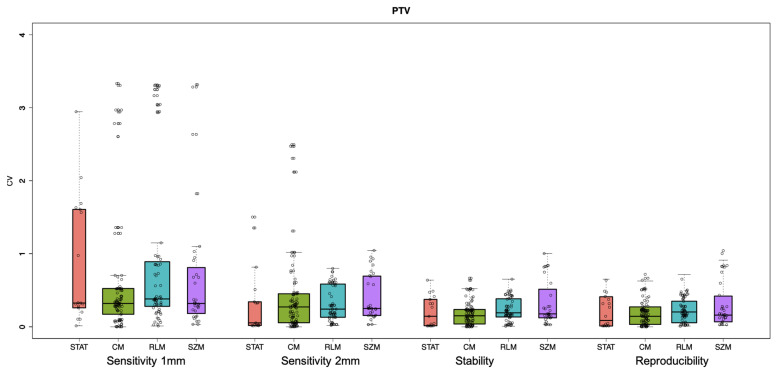
Box plots of the CV values for the sensitivity (1 mm and 2 mm), stability and reproducibility studies grouped for the four different features’ families (STAT, CM, RLM and GSZ) for the PTV.

**Figure 6 cancers-13-03835-f006:**
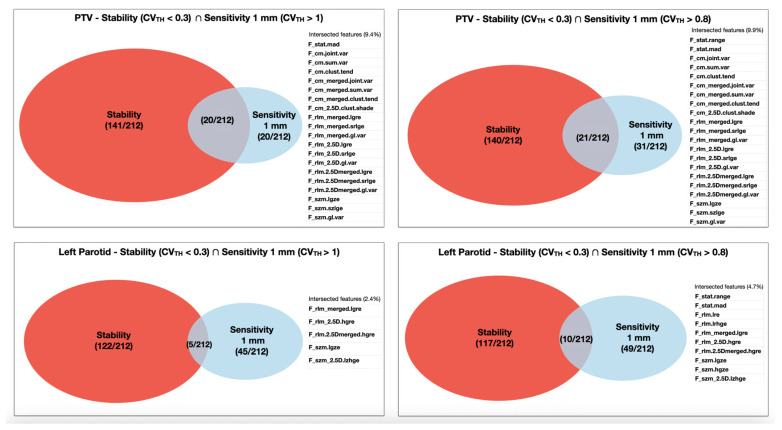
Venn diagrams showing the dosiomic features that are both stable and sensitive between the stability and sensitivity study for the PTV and left parotid. The sensitivity CV threshold was set to two different values: CV_TH_ > 1 and CV_TH_ > 0.8 for the PTV.

**Table 1 cancers-13-03835-t001:** Details of the IMRT protocol used to study the stability of the features when derived from almost equivalent plans: beam setup, dose prescription and constraints for organs at risk.

**Beam settings**	**Gantry Angles**	**Energy**	**Dose Rate**	**Collimator Angles**	**Dose Calculation Algorithm**	**Iteration**
	0°, 40°, 80°, 120°, 160°, 200°, 240°, 280°, 320°	6 MV FF	300 MU/min	15 for all the fields	AAA	At least 700
**Planning objectives**	Upper: Vol(%) = 0, Dose(Gy) = 68, Priority = 140			Lower: Vol(%) = 100, Dose(Gy) = 66, Priority = 140		
**OARs constraints**	**Trachea**	**Parotid L**	**Parotid R**	**PRV SC**	**Spinal canal**	**RING**
	D_mean_ = 49.5 Gy Priority = 80	D_mean_ = 5.0 Gy Priority = 50	D_mean_ = 23.0 Gy Priority = 100	D_max_ = 40.0 Gy Priority = 90	D_max_ = 62.0 Gy Priority = 110	D_max_ = 39.96 Gy Priority = 90

FF = flattening filter; OARs = Organs at Risk; MU = monitor units; D_mean_ = mean dose; D_max_ = max dose; AAA = anisotropic analytical algorithm; Vol = volume; Parotid L = left parotid; Parotid R = right parotid; PRV SC = planning organ-at-risk volume for the spinal canal.

**Table 2 cancers-13-03835-t002:** List of centres that computed the IMRT dose distribution for the reproducibility and stability studies.

Centre_Plan	LINAC
G_1	CLinac
E_1	TrueBeam
E_2	Edge
B_1	TrueBeam
B_2	Edge
D_1	TrueBeam
D_2	Trilogy
A_1	TrueBeam

**Table 3 cancers-13-03835-t003:** List of the eleven plans generated by the centres involved in the study, type of particle, beam energy, delivery technique, kind of Linac, treatment planning system and dose calculation algorithm.

Centres_Plan	Particle	Energy (MV)	Technique	Accelerator Devices	TPS	Dose Calculation Algorithm
A_S1	photon	6 FF	VMAT	TrueBeam. Varian	Eclipse	AAA
B_S2	photon	6 FF	VMAT	TrueBeam. Varian	Eclipse	AAA
C_S3	proton	62.3–226.9 MeV/u	IMPT	Synchrotron (CNAO) [27]	RayStation	MC
D_S4	photon	6 FF	VMAT	TrueBeam. Varian	Eclipse	AAA
E_S5	photon	6 FF	VMAT	TrueBeam. Varian	Eclipse 15.6	Acuros
F_S6	photon	6 FF	VMAT	Synergy. Elekta	Pinnacle	CC
F_S7	photon	6 FF	VMAT	Synergy, Elekta	RayStation	CC
G_S8	photon	6 FF	VMAT	Clinac, Varian	Eclipse	AAA
H_S9	photon	6 FFF	TOMO	Tomotherapy, Accuray	Tomotherapy HT 2.1.6	CC
H_S10	photon	6 FF	DWA	Vero, *Brainlab*-Mitsubishi	Raystation 9B SP2	MC, CC
H_S11	photon	6 FF	VMAT	Trilogy, Varian	Eclipse 15.6	AAA

DWA = dynamic wave arc; TPS = treatment planning system. FF = flattening filter; VMAT = Volumetric Modulated Arc Therapy; IMPT = Intensity Modulated Proton Therapy; TOMO = Tomotherapy; AAA = anisotropic analytical algorithm; MC = MonteCarlo; CC = collapsed cone.

**Table 4 cancers-13-03835-t004:** Dose prescription and constraints to organs at risk employed for the generation of the dose distributions for the sensitivity study.

ROIs	Dose Prescription and Constraints	
PTV	D98% > 95%	V105% < 10%
Spinal canal	D_max_ < 45 Gy	
PRV spinal canal	D_max_ < 45 Gy	
Trachea	D_mean_ < 50 Gy	
Parotids	D_mean_ < 25 Gy	
RING	D_max_ < 95% = 62.7 Gy	

PTV = Planning Target Volume; PRV SC = planning organ-at-risk volume for the spinal canal; D_mean_ = mean dose; D_max_ = maximum dose; D98% = minimum dose to the 98% of the volume; V105% = percent of volume receiving at least 105% of the prescribed dose.

**Table 5 cancers-13-03835-t005:** List of the percentage of common dosiomic features among different studies and different CV thresholds.

ROIs	Repr. (CV_TH_ < 0.3) ∩ Stab. (CV_TH_ < 0.3)	Sens. 1 mm (CV_TH_ > 1) ∩ Sens. 2 mm (CV_TH_ > 1)	Stab. (TH < 0.3) ∩ Sens. 1 mm (TH > 1)	Stab. (CV_TH_ < 0.3) ∩ Sens. 2 mm (CV_TH_ > 1)
PTV	63.2%	5.7%	9.4%	1.9%
Left parotid	49.5%	14.2%	2.4%	1.4%
Right parotid	46.2%	5.7%	9.0%	2.8%
Spinal canal	68.9%	12.7%	3.3%	2.8%
Trachea	58.5%	16.5%	2.8%	2.8%
RING	82.5%	1.4%	0.0%	0.5%

Repr. = Reproducibility (green), Stab. = Stability (red), Sens. 1 mm= Sensitivity 1 mm (light blue), Sens. 2 mm= Sensitivity 2 mm (dark blue).

## Data Availability

The data presented in this study are available on request from the corresponding author. The data are not publicly available due to privacy reasons, according to GDPR.

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
