# Peer review of "A Multicentre Evaluation of Dosiomics Features Reproducibility, Stability and Sensitivity"

_cancers, 2021, doi:10.3390/cancers13153835_

Round 1
Reviewer 1 Report
General comments:
This is a topical paper resulted from a nation-wide collaboration between a large number of RT centres involved in radiomics / dosiomics that aims to evaluate the reproducibility, stability and sensitivity of dosiomic features using head and neck cancer phantom-based treatment planning and analysis.
The paper is generally well written and interesting. Below are my suggestions to further improve the readability of the manuscript:
Introduction: It would help the flow of the paper to add a short paragraph (before starting to present the concept behind dosiomics) on the role of DVHs in treatment planning interpretation / comparison and optimisation, while also mentioning the main shortcoming of DVHs – the lack of spatial dosimetric information. This is a good justification for the development of dosiomics that allow a volumetric assessment of dose distribution with the PTV and OARs.
Introduction – lines 86-87 “Placidi et al. evaluated the robustness of dosiomic signatures across grid resolution and algorithm for dose calculation in a monocentric setting” – and what was found in that study? A few summary sentences should be included here to justify the current, multi-centre study.
In Materials and Methods, the authors state that “Specifically, nine centres have contributed to this analysis” while table 3 is a “list of the 11 centres” – I believe this is a list of 11 treatment plans by the 9 centres. This should be corrected / clarified.
Section 2.2 – line 129 – “IMRT treatments with dose distributions 129 as equivalent as possible” – define / quantify ‘as equivalent as possible’. How were the treatment plans compared?
Discussion – the authors mention several studies that used dosiomics for prediction of clinical outcome. This idea should be continued by expanding on the true value of dosiomics – if we can talk about a ‘true value’ at this stage. How can dosiomics help to improve treatment outcome? What is the benefit of predicting acute toxicities? How would those toxicities be prevented / limited for future patients? Please add a few sentences on this idea.
Where multiple references are used in text in sequential order, please include the first and the last references only. For instance, replace the sequence [5] [6] [7] [8] [9] [10] [11] [12] with [5-12]. Change throughout the manuscript.
Specific comments:
- The title should read ‘A multi-centre evaluation…’ (not A multi-centres…)
- Figure 2. caption – remove from caption the final section: ‘2.3. Plan and dose prescription: different techniques, technologies and TPS.’
- Page 5, line 164 – ‘defined as the smallest absolute…’
Reviewer 2 Report
Generally, it is interesting and worth publishing paper. I have rather few suggestions and questions rather than major comments. Please consider them during review process:
- It is worthy to know why this paper is a multicentre article. Authors should add in the abstract and/or title that work has been done by nine Italian centres.
- In keywords, please add: radiation dosimetry, radiotherapy etc. would be better visible during similar paper searching.
- Figure 1 should be better described (not only in the main text of manuscript), i.e. what means ring etc.
-
Please describe, why dose distribution computation has been performed from eight different centres, not nine?
-
Why authors chose almost similar LINACs in this study?
-
In Table 2 Dmax=max dose. Do you mean total dose as a whole treatment or fractionated?
-
Table 4. Please explain how you calculated the dose prescription and constraints to organs at risk? Is it mean value from 7 centres.
-
In Results part, not sure if that sentence is needed: "This section may be divided into subheadings. It should provide a concise and pre- cise description of the experimental results, their interpretation, as well as the experi- mental conclusions that can be drawn".
